# A Role of Newly Found Auxiliary Site in Phospholipase A_1_ from Thai Banded Tiger Wasp (*Vespa affinis*) in Its Enzymatic Enhancement: In Silico Homology Modeling and Molecular Dynamics Insights

**DOI:** 10.3390/toxins12080510

**Published:** 2020-08-08

**Authors:** Withan Teajaroen, Suphaporn Phimwapi, Jureerut Daduang, Sompong Klaynongsruang, Varomyalin Tipmanee, Sakda Daduang

**Affiliations:** 1Biomedical Sciences Program, Graduate School of Khon Kaen University, Khon Kaen 40002, Thailand; withan_t@kkumail.com; 2Faculty of Science, Khon Kaen University, Khon Kaen 40002, Thailand; psupha@kku.ac.th; 3Centre for Research and Development of Medical Diagnostic Laboratories, Faculty of Associated Medical Sciences, Khon Kaen University, Khon Kaen 40002, Thailand; jurpoo@kku.ac.th; 4Protein and Proteomics Research Center for Commercial and Industrial Purposes (ProCCI), Khon Kaen 40002, Thailand; somkly@kku.ac.th; 5EZ-Mol-Design Laboratory and Department of Biomedical Sciences, Faculty of Medicine, Prince of Songkla University, Songkhla 90110, Thailand; 6Division of Pharmacognosy and Toxicology, Faculty of Pharmaceutical Sciences, Khon Kaen University, Khon Kaen 40002, Thailand

**Keywords:** phospholipase A_1_, Ves a 1, *Vespa affinis*, homology modeling, molecular dynamics

## Abstract

Phospholipase A_1_ from Thai banded tiger wasp *(Vespa affinis)* venom also known as Ves a 1 plays an essential role in fatal vespid allergy. Ves a 1 becomes an important therapeutic target for toxin remedy. However, established Ves a 1 structure or a mechanism of Ves a 1 function were not well documented. This circumstance has prevented efficient design of a potential phospholipase A_1_ inhibitor. In our study, we successfully recruited homology modeling and molecular dynamic (MD) simulation to model Ves a 1 three-dimensional structure. The Ves a 1 structure along with dynamic behaviors were visualized and explained. In addition, we performed molecular docking of Ves a 1 with 1,2-Dimyristoyl-sn-glycero-3-phosphorylcholine (DMPC) lipid to assess a possible lipid binding site. Interestingly, molecular docking predicted another lipid binding region apart from its corresponding catalytic site, suggesting an auxiliary role of the alternative site at the Ves a 1 surface. The new molecular mechanism related to the surface lipid binding site (auxiliary site) provided better understanding of how phospholipase A_1_ structure facilitates its enzymatic function. This auxiliary site, conserved among Hymenoptera species as well as some mammalian lipases, could be a guide for interaction-based design of a novel phospholipase A_1_ inhibitor.

## 1. Introduction

Thai banded tiger wasp (*Vespa affinis*) is a dangerous wasp species commonly found in many regions of Asia [1]. The secreted toxic components from venom glands are vital to wasp survival in protection against enemies and in prey hunting [2]. The toxin leads to local allergic reactions; acute renal failure; dysfunction of organs, including hemolytic and hepatitis dysfunction; and even fatal allergy [3,4]. In addition, vespid venom component composes three major proteins, including phospholipase A_1_, protease, and hyaluronidase [5,6,7]. The wasp venom phospholipase A_1_ (Ves a 1) becomes of interest in the interruption of victim cell membrane aggregation. The vespid Ves a 1 venom belongs to subgroup number eight of the pancreatic lipase gene family. An enzymatic function focuses on the hydrolysis of the sn-1 position of phospholipids [8]. 

Ves a 1 exhibited similarity of conserved regions to human pancreatic lipase (HPL) and pancreatic lipase-related protein 2 (PLRP2). Moreover, these regions of enzymes contain the catalytic triad of Ser-His-Asp (S-H-D) [9]. The N-terminal domain consisted of an α/β hydrolase fold and a sequence of Gly-X-Ser-X-Gly (G-X-S-G-X) preserved in lipase and esterase. However, three tertiary domains, which are the lid domain, β5, and β9 loop, are directly associated with a catalytic role. Lid domain and β9 loop play a role in substrate selection and recognition of the sn-1 position, while the β5 loop is a hole for the phospholipid carbonyl group. These components enhance the substrate accessibility to the enzyme catalytic site. Moreover, the three-dimensional (3D) structure from black-bellied hornet *(Vespa basalis*) phospholipase A_1_ was characterized in the form of a crystal structure and provided useful atomistic information, and the mechanism of phospholipid binding of phospholipase A1 was proposed [10]. The roles of α5, β5, and β9 loops were also reported for a function enhancement [11]. Ves a 1 also possessed a similar catalytic site to PLA1 from *V. basalis* as well as an open conformation. However, the mechanism of how PLA1 encounters the phospholipid bilayer remains unknown. 

In this study, we performed the 3D structural modeling of Ves a 1 so that we can investigate its structural behaviors and lipid-binding characteristic. This study could lead to a comprehensive role of amino acid difference in Ves a 1 compared with the PLA1 from *V. basalis* in terms of lipid selectivity and recognition. Besides, a proposed Ves a 1 model and its interaction with a phospholipid molecule could also provide valuable information for finding the new PLA1 inhibitor for the purpose of therapeutic application and vespid toxin-associated allergy.

## 2. Results

### 2.1. Ves a 1 Modeling

In this study, the sequence of phospholipase A_1_ from *V. affinis* venom (Ves a 1) was compared to the phospholipase A_1_ from black-bellied hornet (*V. basalis*) venom (Uniprot KB accession number A0A0M3KKW3) (PA1-V.bas) because PA1-V.bas is the only available 3D crystal structure of vespid phospholipase A1. Sequence alignment between these two proteins yielded an identity percentage of more than 90%, shown in Figure 1. The identity came from 297 identical residues (92.69%), 18 strongly similar residues (5.98%), and 4 different residues (1.33%). Moreover, the secondary structure and the modeled tertiary structure of the *V. affinis* phospholipase A_1_ showed similarity to its *V.basalis* Ves a 1 counterpart (Figure 2). The comparison gave a hint that the *V. basalis* phospholipase A_1_ can be well represented as a molecular template for Ves a 1. However, an insertion of Tyr209 (Y209) residue on template sequences and the replacement of 16 residues according to Ves a 1 sequence were applied onto the template sequence to perform protein fold recognition modeling. The predicted 3D structure was confirmed with the aforementioned secondary structure prediction. Ramachandran plot was also carried out, and only 0.4% (1 residue) on the disallowed region was observed (Figure 3). These results led to a suitably predicted Ves a 1 structure; however, molecular dynamics simulation was still required for structure refinement and conformational study of Ves a 1.

### 2.2. Molecular Dynamic Simulation and Molecular Docking

The conformational stability of modeled Ves a 1 was investigated through steady root mean square displacement (RMSD) of about 1.45 Å, where an equilibrium phase was observed at 100 nanoseconds (ns) (Figure 4). Ves a 1 consists of 6 disulfide bridges which could enhance Ves a 1 stability [12]. MD simulation was performed for the protein folding base on a hypothesis of whether Ves a 1 requires a similar conformation to preserve catalytic feature [10,12]. Besides, a steady radius of gyration (Rg) for Ves a 1 was observed versus simulation time to visualize the protein compactness. These implied that 200-ns simulation sufficed for a Ves a 1 conformational study. 

Additionally, the secondary structure of each residue in Ves a 1 residue was plotted against simulation time to track the conformational changes of the 3D structure (Figure 5). The secondary structure was obtained from 500 snapshots of the 200 ns MD trajectory. Figure 5 illustrated that the β9 loop and the lid domain displayed various secondary structures after 20 ns in dynamic trajectories. The α9 region was flexible helices. This result indicated that catalytic regions of Ves a 1 adopt a resilient structure during folding. In addition, Ves a 1 surface activation was examined moving from a helical shape into a coil; these results may confirm the behavior of Ves a 1 for substrate accessibility [12].

Finally, profiles of time-averaged root mean square fluctuation (RMSF) plots illustrated Ves a 1 residue movement (Figure 6). The fluctuation of regions Glu89-Ala96 (E89-A96) (cover α9 region), Gln216-Cys228 (Q216-C228), and Ser253-Asn265 (S253-N265) (cover lid domain and β9 loop region) were investigated. The fluctuation was approximately 2–2.5 Å, indicating that the amino acid regions of the surface interacting site remained agile during the MD simulation [8,10,12,13].

### 2.3. Ves a 1 Auxiliary Binding Site Identification and Interaction with Phospholipid 

Ves a 1 function was commonly known for phospholipid hydrolysis into fatty acid and lisophospholipids. Various lipid molecules can then be considered as a potential representative to analyze Ves a 1 binding towards a phospholipid molecule. However, in this study, 1,2-Dimyristoyl-sn-glycero-3-phosphorylcholine (DMPC) was chosen as a lipid representative since DMPC could well represent the phospholipid bilayer in both computational and experimental studies [14,15,16,17,18]. In this study, the lipid was docked into the 200-ns snapshot of Ves a 1 and the binding energy as well as conformation was obtained using SwissDock. The binding residues were predicted for interfacial interaction on the protein surface, displayed in Figure 7.

The DMPC bound the Ves a 1 catalytic site, corresponding to a previous report [10,11,12], with a binding energy (ΔG) of −8.67 kcal mol^−1^. Interestingly, the best docked conformation was instead found with a ΔG of −9.77 kcal mol^−1^, which could be an alternative lipid site of Ves a 1. The Ves a 1 interacted with DMPC using Asn195 (N195), Leu200 (L200), Tyr209 (Y209), Asn211 (N211), Asn212 (N212), Gly213 (G213), Tyr214 (Y214), Asn215 (N215), Pro217 (P217), Gln252 (Q252), Lys254 (K254), and Asn255 (N255) (Figure 8). The Asn211 distinctively contributed the binding interaction through hydrogen bonding with a phosphate group in DMPC where other amino acid residues play a role via hydrophobic interactions. This suggested that Ves a 1 would bind the phospholipid bilayer using the preferred alternative site first, prior to a catalytic site.

## 3. Discussion

Ves a 1 is major component of Thai wasp (*V. affinis*) venom and was previously reported for interesting biochemical and bioactivity characteristics [8]. The unavailable 3D structure of Ves a 1 causes a lack of understanding in molecular function. In this study, we have tried to model the 3D structure using Ves a 1 from a black-bellied hornet (*V. basalis*) [10] as a template because of its high protein sequence similarity. The 3D homology modeling combined with MD simulation successfully yielded a stable and reasonable 3D structure of Ves a 1 from *V. affinis* even though about 22 amino acids were different with an additional Tyr209 compared to the *V. basalis* phospholipase A_1_. The molecular docking study provided evidence of the potential binding modes of DMPC with the active site, but the surface site predicted significantly better binding energy. Looking into this surface site at the Ves a 1 surface, the region of Ves a 1 consisted of hydrophobic residues representing potential binding with an acryl chain of phospholipid [19]. The hydrogen bonding due to Asn211 (N211) also contributed to the lipid preference. This site was also surprisingly rigid as the amino acids in the site showed a low fluctuation (RMSF less than 1.0 Å). Furthermore, to be noted, many amino acids at the Ves a 1 surface were located between flexible Q216-C228 (lid domain) and S253-N265 (β9 loop region). Therefore, we have proposed an explanation of Ves a 1 mechanistic action and how this newly found potential site could benefit enzymatic activity.

The lipid-preferred surface site in our Ves a 1 study, defined as an auxiliary site, may not act as a competitive catalytic site, according to the sequence comparison among the insect PLA1s [7,8,9,10,11,20]. Nonetheless, it could play a supporting role in Ves a 1 function. Our finding thus came up with the newly proposed mechanism of how PLA1 hydrolyzes the lipid molecule in the bilayer (Figure 9). Previously, the mechanism suggested that PLA1 directly goes into the bilayer membrane and performs a catalytic using α5 loop to induce the lipid into the catalytic site (Figure 9a) [10]. However, this mechanism remained quite difficult as PLA1 had to functionally rely on a chance protein–lipid encounter where the soluble PLA1 in the cytosol was very mobile. To increase the encounter chance, the surface PLA1 auxiliary site became important for the protein to attach the membrane surface first and to make PLA1 structure more static. The “fixed” PLA1 structure can enhance protein–lipid encounter since the protein was located on the membrane surface and the auxiliary site acted as a fulcrum point for PLA1 to facilitate its catalytic site with the support of an α5 loop [10,11] in binding the target lipid molecule (Figure 9b). Our proposed idea could fulfill the gap in the PLA1 mechanism with substrate–enzyme complex formation prior to the enzyme functioning.

To investigate the atomistic information more deeply, the sequence alignment of the Ves a 1 catalytic site among other vespid phospholipase A_1_ as well as other mammalian lipases such as from human and rats was performed. Furthermore, the newly found auxiliary site was also observed to see whether these two sites were conserved in other organisms. The protein sequences were retrieved from the Uniprot KB database with an addressed Uniprot KB accession code. The Ves a 1 sequence was aligned with five phospholipase A_1_ sequences from *V. basalis* (A0A0M3KKW3), *V. crabro* (P0CH87), *V. velutina* (C0HLL3), *V. tropica* (P0DPT0), and phospholipase A1 (Q06478) from *Vespula maculata*. Five additional mammalian lipases were selected for the study such as human phospholipase A1 member A (Q53H76), human hepatic triacylglycerol lipase (P11150), rat hepatic triacylglycerol lipase (P07867), pancreatic lipase-related protein 2 from *Cavia porcellus* (Guinea pig) (P81139), and human lipoprotein lipase (P06858). The sequence alignment was then illustrated in Figure 10.

A catalytic site (G-X-S-X-G) region (or more precisely G-X-S-L-G-A) was found completely conserved in all selected species (vespid, rat, and human) as expected, leading to an indication that the catalytic site is vital for the lipase function. Interestingly, our proposed auxiliary site (F-Y-M-N-G-Y-N-Q-P-G-C) became conserved in other vespid phospholipase A_1_ and mammalian lipases. We speculated the G-X-P-G-C-G domain could offer hydrophobic interaction for a phospholipid molecule, whereas Asn211 was identically noticed in all observed species to form a hydrogen bond with a phospholipid molecule. As mentioned, Asn211 and the auxiliary site could thus play as an unknown key role in the lipase functional mechanism, not only in vespid phospholipase A_1_ but also in some mammalian lipases. The auxiliary site role in the lipase function can be the next challenge for a behavioral prediction of phospholipase A_1_.

In the same way, an auxiliary site would also be the alternative target for PLA1 inhibitor development. Compared with other phospholipase classes, the known PLA1 inhibitor is not commonly reported. The main strategy of PLA1 inhibition is to design a lipid mimetic compound or phospholipid bilayer derivatives [21,22] to compete with a substrate. The idea of the auxiliary site inhibition would be of interest because the site is a solvent-accessible surface. Besides, due to its larger site area than the catalytic site, it would be easier for a various design of inhibitor shapes apart from the linear-liked phospholipid shape. In conclusion, the Ves a 1 newly found auxiliary site could pave the way for an alternative regime for PLA1-associated therapeutic application.

## 4. Materials and Methods 

Material and method included protein data bank (PDB) structure file and amino acid sequence, web-based Bioinformatics tools and resources, MD simulation protocol, docking protocol, and visualizing software.

### 4.1. Phospholipase A_1_ Structure Preparation

Due to unavailability of phospholipase A1 from the *V. affinis* (Ves a 1) three-dimensional protein in the structure database, homology modeling followed by MD simulation was a selected tool to circumvent this problem and to illustrate the role of Ves a 1 from *V. affinis* [8]. In this study, we employed a primary sequence of Ves a 1 to draft a tentative three-dimensional structure. Furthermore, an MD simulation was carried out to refine the predicted conformation as well as to investigate structural stability and dynamic folding of the protein conformation.

All protein sequences were obtained from Protein Knowledgebase (Uniprot KB) database (www.uniprot.org/uniprot/). All sequence alignment procedures of the Ves a 1 protein with other phospholipase A_1_ and mammalian lipases were carried out with the Clustal Omega Program implemented in the Uniprot KB database. The structure prediction was launched with a target sequence (Uniprot KB accession code P0DMB4). The Ves a 1 template was searched and compared using the SWISS-MODEL library [23], and the crystal structure of phospholipase A1 from *V. basalis* venom (PDB identification code 4QNN) [10] was then taken as a molecular template from RSCB PDB database [24]. For each (target and template), a secondary structure was independently predicted using two Bioinformatics tools, PSIPRED Protein Sequence Analysis Workbench [25,26] and NetSurfP version 1.1 [27], to inspect the consistency of the predicted structure. The target structure was finally chosen from 125 possible structures corresponding to the predicted secondary structure. 

Since the Ves a 1 template structure was taken from a different species, some amino acids based on Ves a 1 from Thai banded tiger wasp venom were corrected corresponding to Ves a 1 sequence from *V. affinis*. First of all, Tyr209 was added using Phyre2 server [28], yielding 301 amino acids. The 21 mutagenized points were generated using the Rosetta Backrub server [29] to be Gln29, Gln33, Thr56, Thr58, Ala59, Glu67, Leu70, Asn89, Ala92, Val94, Lys95, Gln123, Arg154, Gln174, Glu175, Gln178, Glu182, Asn186, Ile223, Arg242, and Asn255. The selection of the predicted structure was performed using the best result from 50 possible structures based on the highest prediction score. The structure was also checked for the disallowed conformation by Ramachandran plot from PROCHECK [30]. The best predicted structure was used as a starting coordinate for MD simulation.

### 4.2. Molecular Dynamics Simulation of Ves a 1 and Molecular Docking of a DMPC-Ves a 1 Complex 

The protein in a sodium chloride (NaCl) solution at 37 °C was later prepared so as to mimic the in vivo-like condition so that the molecular conformation could be reasonably illustrated. The constructed Ves a 1 structure was solvated in the transferable intermolecular potential with 3 points (TIP3P) water rectangular box with a distance of 12 Å from the protein surface, yielding 15,846 water molecules, using the Assisted Model Building with Energy Refinement (AMBER16) package [31]. The protein charge was then neutralized using 9 chloride ions (Cl^−^), and 43 NaCl pairs were then added into the system to create the equivalent concentration of 0.15 mol/L NaCl solution. Each protonation state of all ionizable amino acid side chains was considered at pH 7. The system was therefore modeled using an AMBER10 force field in AMBER 16 package. Prior to MD simulation, the system was energy minimized for 3000 steps to remove unusual inter-atomic contacts through the steepest descent method (2000 steps), followed by the conjugate gradient method (1000 steps). 

For MD simulation, the system was equilibrated by a canonical (constant number (N), volume (V), and temperature (T); (NVT)) ensemble for 600 picoseconds, where the protein was positionally restrained using force constants of 250, 150, 100, 50, 20, and 10 kcal mol^−1^ Å^−2^ for each 100 picosecond simulation. A time step of 1 femtosecond (fs) was applied in each NVT simulation with Langevin dynamics [32] to handle a temperature of 310 K (37˚C). After 600 ps-NVT simulation, the system was changed into an isobar/isothermal (constant number (N), pressure (P), and temperature (T); NPT) ensemble. The Berendsen algorithm [33] was applied to regulate both a temperature of 310 K and a pressure of 1 atm (1.013 Bar). 

The investigation of stability and time-dependent behavior was analyzed from an MD trajectory. The first 160-ns NPT simulation was omitted, and the 200 snapshots from the last 40 ns were taken for analysis. The system stability was evaluated from the root mean square distance (RMSD) in angstrom unit (Å) from Visual Molecular Dynamics (VMD) package, Version 1.9.1 [34]. The flexibility was reflected via the root mean square fluctuation (RMSF) in Å, calculated from using cpptraj in the AmberTools20 package [35]. The molecular docking was performed on Ves a 1 structure and a 1,2-Dimyristoyl-sn-glycero-3-phosphorylcholine (DMPC) using SwissDock [36] with the default docking parameter. The DMPC structure was obtained from PubChem database, PubChem CID: 5459377. The lowest binding energy of binding mode and the protein–lipid schematic interaction of DMPC with Ves a 1 lipid binding site were analyzed using Biovia Discovery Studio visualizer version 20.10 [37]. The DMPC structure was drawn using ACD/ChemSketch (Freeware) package, Version 2019.2.2 [38]. 

## Figures and Tables

**Figure 1 toxins-12-00510-f001:**
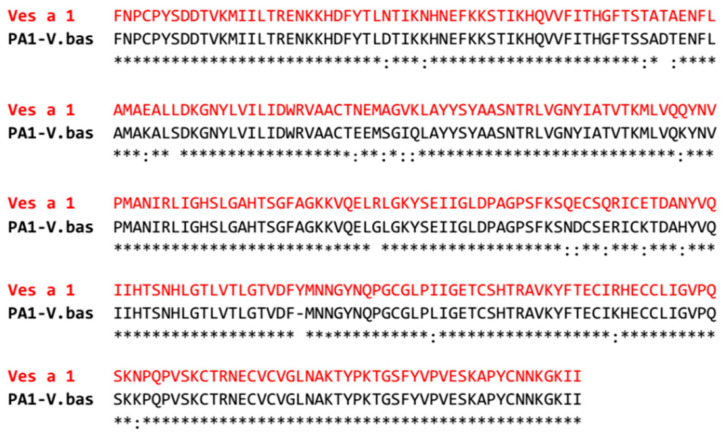
Schematic of protein sequence alignment between and Ves a 1 from *V. affinis* venom (Ves a 1) (red) and Ves a 1 from (*V. basalis*) venom (PA1-V.bas) (black): Ves a 1 and PA1-V.bas are based on the sequences with Uniprot KB accession number P0DMB4 and number A0A0M3KKW3, respectively. The alignment data illustrated 279 identical (*), 18 strongly similar (:), and 4 different amino acids.

**Figure 2 toxins-12-00510-f002:**
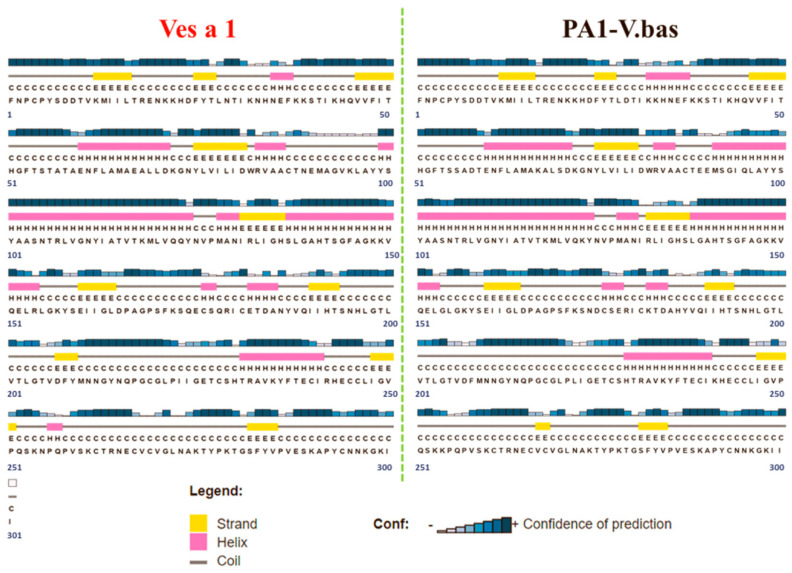
Predicted secondary structures of Ves a 1 from *V. affinis* venom (Ves a 1) (red) and Ves a 1 from (*V. basalis*) venom (PA1-V.bas) (black), respectively: The H and pink rectangle represent a helical structure. The E and yellow rectangle represent a sheet structure. The C and black line represent a coil structure. The height of the blue bar represents the level of confidence in the predicted structure. The number indicates the amino acid order.

**Figure 3 toxins-12-00510-f003:**
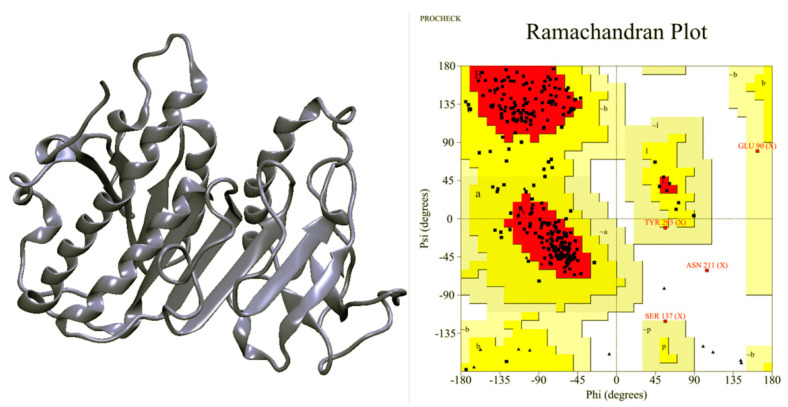
Predicted three-dimensional structure of *V. affinis* phospholipase A_1_ (Ves a 1): The structure was predicted from the protein fold recognition method by adding a 209^th^ tyrosine (Y209) into a sequence template of *V. basalis* phospholipase A_1_. The Ramachandran plot-predicted Ves a 1 phospholipase A_1_ structure contained less than 1% of the disallowed region, which well indicated the validity of the predicted structure.

**Figure 4 toxins-12-00510-f004:**
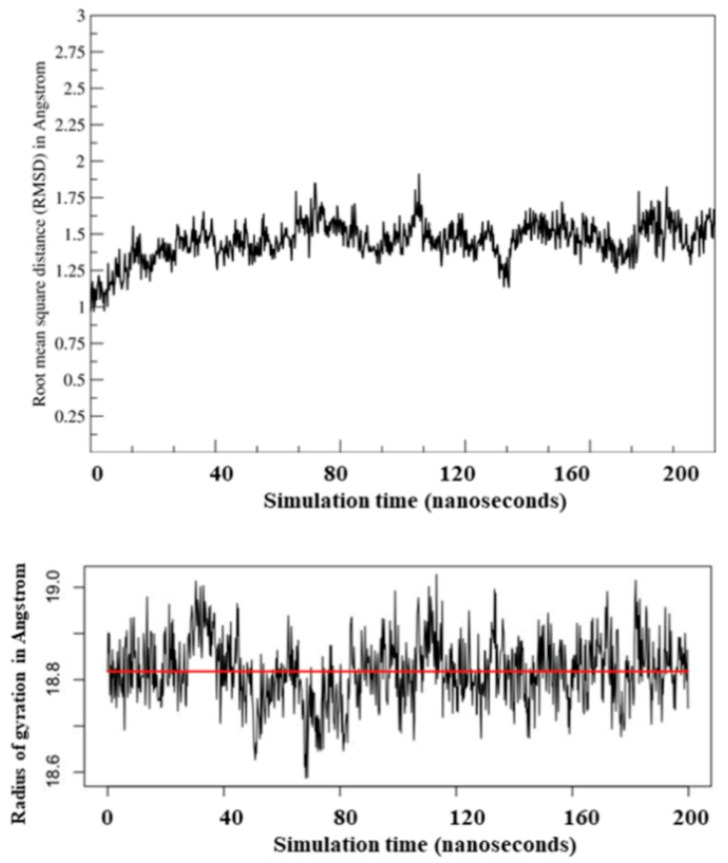
Root mean square distance (RMSD) and radius of gyration (Rg) trajectory plots from phospholipase A_1_ (Ves a 1) molecular dynamic (MD) simulation: The system showed steady RMSD throughout 200-ns simulation with an average RMSD of 1.45 Å and an average Rg of 18.8 Å.

**Figure 5 toxins-12-00510-f005:**
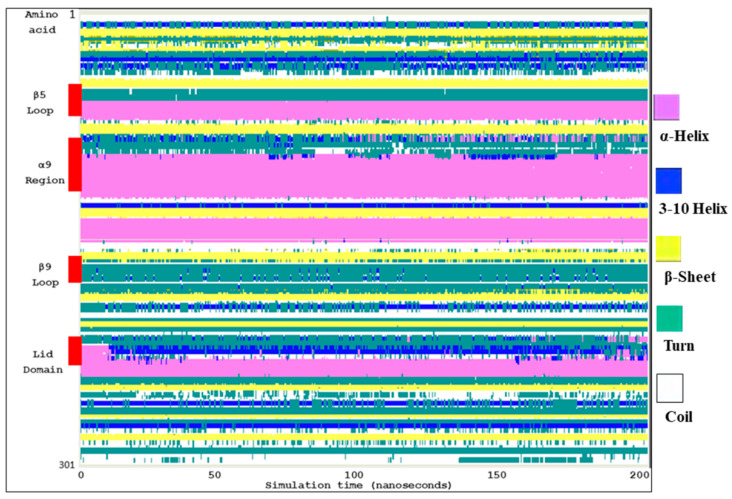
Secondary structure of Ves a 1 (*V. affinis*) from molecular dynamics simulation: The secondary structure pattern was plotted against the simulation time (0–200 ns). The red bar illustrated that the amino acid sequence consists of interfacial activation. The color labels the secondary structure type.

**Figure 6 toxins-12-00510-f006:**
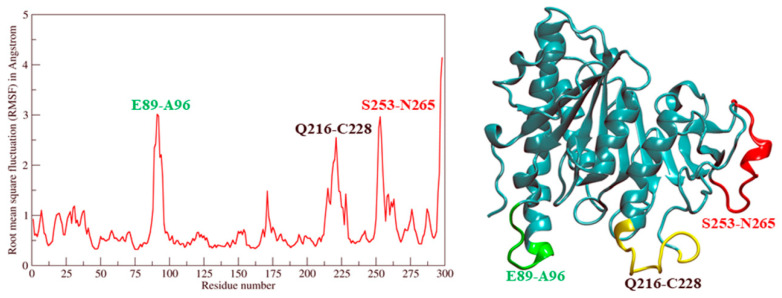
Root mean square fluctuation (RMSF) of α-carbon atoms in Ves a 1 MD trajectory: The distinct flexible components included E89-A96 (α9 region), Q216-C228, and S253-N265 (cover lid domain and β9 loop region).

**Figure 7 toxins-12-00510-f007:**
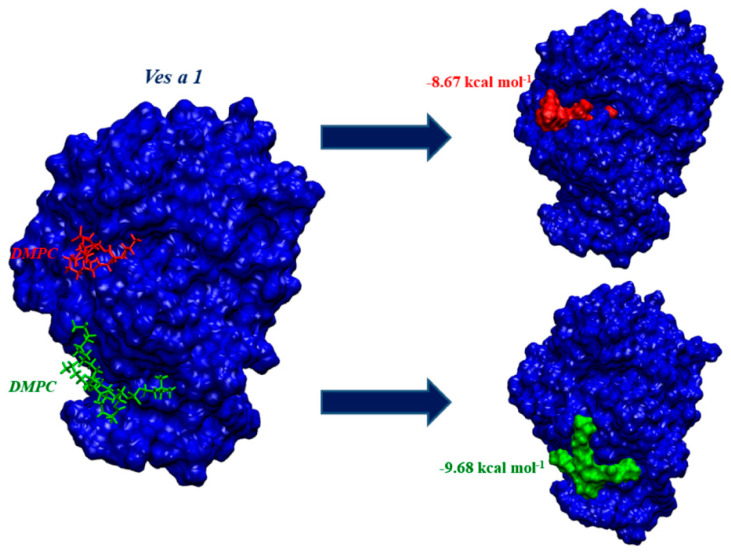
Predicted 1,2-Dimyristoyl-sn-glycero-3-phosphorylcholine (DMPC) target site towards the modeled Ves a 1: The two sites shown were DMPC bound with Ves a 1 at its catalytic site (red) with the binding energy of −8.67 kcal mol^−1^ and DMPC bound with Ves a 1 at its alternative auxiliary site (green) with the lower binding energy, −9.77 kcal mol^−1^.

**Figure 8 toxins-12-00510-f008:**
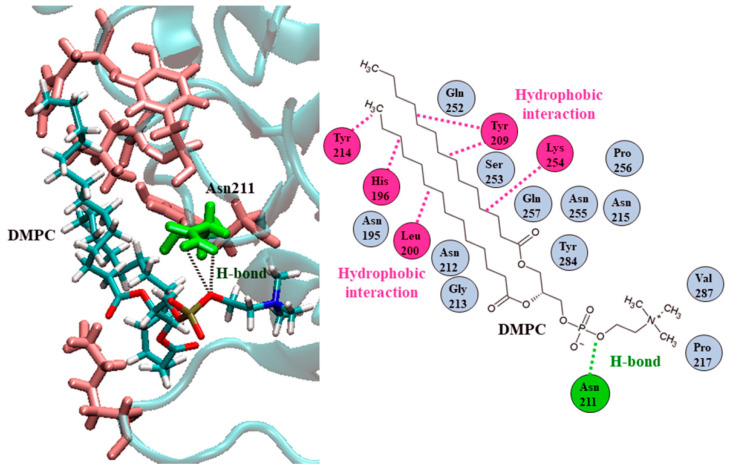
Lipid alternative Ves a 1 binding site (auxiliary site) predicted from a molecular docking study: Asn211 or N211 (green) formed hydrogen bonds (black and green dash lines) with a phosphate group in DMPC. Moreover, hydrophobic interactions (pink dash lines) play a role in DMPC binding with a Ves a 1 alternative site.

**Figure 9 toxins-12-00510-f009:**
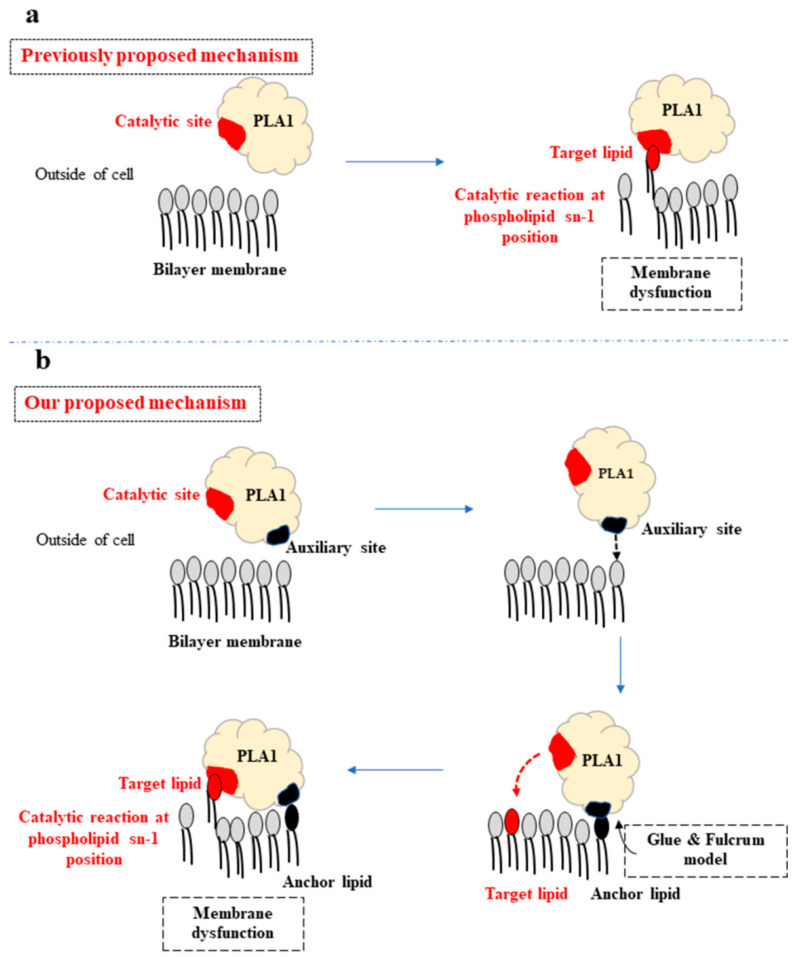
Proposed mechanism of Ves a 1 catalytic action associated with an auxiliary site: (**a**) the previously proposed mechanism suggested that PLA1 in cytosol (outside the cell) went into the membrane directly and exploited its catalytic site (red area) bound the lipid molecule. (**b**) Our proposed mechanism suggested that PLA1 first used the auxiliary site (black area) to stick to a lipid molecule on the membrane surface. Then, the attached lipid molecule (black headed lipid) acted as a fulcrum point for PLA1 to approach its catalytic site towards a nearby target lipid molecule (red head lipid), leading to membrane dysfunction.

**Figure 10 toxins-12-00510-f010:**
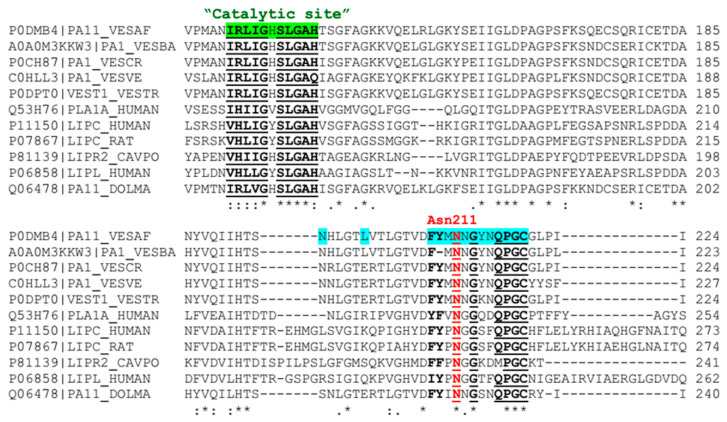
Sequence alignment of Ves a 1 catalytic and auxiliary sites with other lipases: the Ves a 1 sequence from *V.affinis* (PA11_VESAF) was aligned with the other five phospholipase A_1_ (PA1_VESBA, PA1_VESCR, PA1_VESVE, VEST1_VESTR, and PA11_DOLMA8) and five mammalian lipases (PLA1A_HUMAN, LIPC_HUMAN, LIPC_RAT, LIPR2_CAVPO, and LIPL_HUMAN). The corresponding Uniprot KB accession code is listed before the sequence name; for example, P0DMB4|PA11_VESAF indicates a Ves a 1 sequence for which the Uniprot KB accession code is P0DMB4. The green and light blue labeled sites indicate a catalytic site and an auxiliary site, respectively. The red N showed the Asn211 position which was responsible for binding a phospholipid molecule at an auxiliary site. The asterisk (*) and colon (:) point out the identical and conserved regions among species.

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
