# Peer review of "A Role of Newly Found Auxiliary Site in Phospholipase A1 from Thai Banded Tiger Wasp (Vespa affinis) in Its Enzymatic Enhancement: In Silico Homology Modeling and Molecular Dynamics Insights"

_toxins, 2020, doi:10.3390/toxins12080510_

Round 1
Reviewer 1 Report
The manuscript “Toxins-868829” has carried out in silicon analysis to study homology modeling, molecule dynamic simulation and docking of Vespid phospholipase A 1 (Ves a 1) in Vespa affinis. The modeling of Ves a A1 was described in detail. In addition, the author (s) also reported the predicted active site of the Ves a 1 and they showed a predicted binding region between Ves a 1 and DMPC. However, this manuscript still requires to be improved as commented below:
- To be novel from previous studies about Ves a 1 and other phospholipase A1 in insects, it is good to know function of the predicted binding site between Ves a 1 and DMPC by in vitro enzymatic activity assay of Ves a 1 after mutating binding residues.
- The authors need to improve resolution of their figures (Fig. 1, 2, 3a and 6, for instance). The figures were presented unclear so it is hard to understand what they wanted to show.
- Lastly, the manuscript should go for language editing since it had problems in term of English language that cause difficulty in reading the manuscript.
Author Response
Dear Reviewer 1
We would like to respond the comment from you as followings:
1. To be novel from previous studies about Ves a 1 and other phospholipase A1 in insects, it is good to know function of the predicted binding site between Ves a 1 and DMPC by in vitro enzymatic activity assay of Ves a 1 after mutating binding residues.
Response:
Thanks for addressing this point, we don't perform any addtional in vitro experiment the new predicted binding site does not relate to the direct catalytic process of lipid hydrolysis. Instead we presented the auxilliary site of V. affinis phospholipase A1 (Ves a 1) as a key role for bilayer membrane attachment of Ves a 1 and facilitate the Ves a 1 function. We also propose the "new" possible mechanism of Ves a1 in the lipid hydrolysis associated with the auxilliary site role in our finding. In addition, this new found site could pave a way for the novel phospholipase A1 inhibitor to target the alternative auxilliary site, in a case of allergen and other therapeutical application in toxin remedy.
2. The authors need to improve resolution of their figures (Fig. 1, 2, 3a and 6, for instance). The figures were presented unclear so it is hard to understand what they wanted to show.
Response:
All figures in the manuscript were improved and rearranged to clearly inform the reader and make a focus on the related content. All figures complied a required publication standard resolution (300 dpi), and the figure legends and captions were rewritten to point out the details illustrated in the figure.
3. Lastly, the manuscript should go for language editing since it had problems in term of English language that cause difficulty in reading the manuscript
Response:
We are terribly sorry for the poor writing and we strongly agreed at your point. The manuscript was now whole rewritten and grammatically checked so the format should be in the understandable English and meet the publication standard.
We are grateful for your constructive comments to fulfill the missing point in our manuscript.
Kind regards,
Reviewer 2 Report
The manuscript entitled " In silico prediction for homology modeling and molecular dynamic simulation of phospholipase A1 (Ves a 1) from Thai banded tiger wasp (Vespa affinis) venom" describes the high-resolution protein 3D structure, MD simulation and docking simulation of Ves a 1.
First of all, as far as I checked in abstract and introduction sections, I found many grammatical mistakes. For instance, the sentence of “to evaluate the commercialization……” in lines 7 and 8 is an incomplete sentence. And the sentence of “however, functional of enzyme is…….”in lines 33 and 34 might be inappropriate and does not make sense. Such mistakes are not limited in abstract and introduction, and I found too many mistakes in whole manuscript to point out. Due to the many grammatical mistakes, my impression of this manuscript is not so good. Before submission, the authors should brush up the manuscript more carefully and completely.
Because high similarity with template structure, most of the analyses performed in this study use the programs on the web site, and might not require any special knowledge or technique very much, this study seems to be lacking novelty. I recommend you should more emphasize the novelties of this study in discussion.
Author Response
Dear Reviewer 2,
I took a responsibility to handle with your comments as followings.
============
1.“The manuscript entitled " In silico prediction for homology modeling and molecular dynamic simulation of phospholipase A1 (Ves a 1) from Thai banded tiger wasp (Vespa affinis) venom" describes the high-resolution protein 3D structure, MD simulation and docking simulation of Ves a 1.”
Response: We have changed the manuscript title into “A role of newly found auxiliary site in phospholipase A1 from Thai banded tiger wasp (Vespa affinis) in its enzymatic enhancement: in silico homology modeling and molecular dynamics insights” so that a reader can understand briefly for the what our finding is.
============
2. “First of all, as far as I checked in abstract and introduction sections, I found many grammatical mistakes. For instance, the sentence of “to evaluate the commercialization……” in lines 7 and 8 is an incomplete sentence. And the sentence of “however, functional of enzyme is…….”in lines 33 and 34 might be inappropriate and does not make sense. Such mistakes are not limited in abstract and introduction, and I found too many mistakes in whole manuscript to point out. Due to the many grammatical mistakes, my impression of this manuscript is not so good. Before submission, the authors should brush up the manuscript more carefully and completely.“
Response: I am so sorry for such a careless and unimpressive manuscript with the poor English and mistakes. I do agree with your point. In this rewritten manuscript, we have improved the writing and grammatical-checking. Also we have get rid of many mistakes not only the content but also the figure resolution to meet the publication standard. We can ensure that the revised manuscript can be read in more understandable English.
============
3.”Because high similarity with template structure, most of the analyses performed in this study use the programs on the web site, and might not require any special knowledge or technique very much, this study seems to be lacking novelty. I recommend you should more emphasize the novelties of this study in discussion.”
Response: Thanks for addressing this point. We have used web-based website to generate some general prediction however, the core technique we used is molecular dynamics simulation to track the dynamics behaviours as well as the reasonable structural refinement. Even though we performed the molecular docking online, the analysis was performed manually as well as raising the interaction issue and conclusion.
To answer the concern of novelty, we read back the first draft of manuscript and we agree we lack of novelty. Therefore we have added more novelty details in many parts of the revised manuscript which are
-In an abstract from the line 31 to 34:
“The new molecular mechanism related to the surface lipid binding site (auxiliary site) was then proposed to provide the better understanding of how phospholipase A1 structure facilitates its enzymatic function. Besides, the auxiliary site at Ves a 1 surface could be a guide for interaction-based design of a novel phospholipase A1 inhibitor.”
-In a discussion section from the line 173 to 186 giving general details of our work we have done
-In a discussion section from the line 187 to 194 emphasized our novelty:
“Moreover, an auxiliary site could also be the alternative target for PLA1 inhibitor development. Compared with other phospholipase classes, known PLA1 inhibitor is not commonly reported. The main strategy of PLA1 inhibition is to design a lipid mimetic compound or phospholipid bilayer derivatives [16,17] so as to compete with a substrate. The idea of the auxiliary site inhibition would be of interest because the site is solvent-accessible surface. Besides, due to its larger site area than the catalytic site, it would be easier to various shape of inhibitor rather than the linear-liked phospholipid shape. In conclusion, the Ves a 1 newly found auxiliary site could pave the way for the alternative regime for PLA1-associated therapeutic application.”
In addition, the figure 10 to propose the new mechanism of phospholipase A1 in catalytic process regarding the newly found alternative site and its role on enzyme activity enhancement.
============
We have put an effort according to your constructive comment of this work in the revised manuscript.
Thanks very much for your comments and time to read our manuscript.
Sincerely yours,
Round 2
Reviewer 1 Report
The manuscript “A role of newly found auxiliary site in phospholipase A1 from Thai banded tiger wasp (Vespa affinis) in its enzymatic enhancement: in silico homology modeling and molecular dynamics insights” has been improved after revising. However, it still needs some minor revision.
- Figure 1:
+ Why do the author use g4QNNx0 only to align with 567011171? Should the author also align 567011171 with the other PLA1 besides g4QNNx0?
+ The author should use g4QNNx0 consistently through all figures and in the text. They use 4QNN or Ves a 1 sometimes so it makes confusing.
+ Should the author simply use the name of “567011171”? And it is wrongly used in the legend of Fig. 1 as “56701171”.
I suggest the author change “g4QNNx0” and “567011171” to a general name which is Ves a 1 and indicate they are in different species.
- Figure 2: Resolution needs to be improved.
- Figure 3: Resolution needs to be improved.
- Figure 6: Resolution needs to be improved. And again, 4QNN and Ves a 1 require to be used consistently in the legend and in the figure.
- Is it Figure 8 before Figure 9? Should it have Figure number?
6. Is the catalytic site of Ves a 1 (Vespa affinis ) conserved with other PLA1? Why do author choose DMPC as a possible binding candidate? Can Ves a 1 of Vespa affinis also hydrolyze other lipids?Please discuss about this issue.
Author Response
Dear Reviewer,
Thanks for your comment. We would like to respond the comment given herein separately in each issue.
- Figure 1:
“+ Why do the author use g4QNNx0 only to align with 567011171? Should the author also align 567011171 with the other PLA1 besides g4QNNx0?
+ The author should use g4QNNx0 consistently through all figures and in the text. They use 4QNN or Ves a 1 sometimes so it makes confusing.
+ Should the author simply use the name of “567011171”? And it is wrongly used in the legend of Fig. 1 as “56701171”.
I suggest the author change “g4QNNx0” and “567011171” to a general name which is Ves a 1 and indicate they are in different species”
Response: Thanks for your comment. We do agree that it could make readers confuse. Therefore, in figure 1, we have changed the name of g4QNNx0 and 567011171 into more general name. To eliminate reader’s confusion, the 567011171 is changed into Ves a 1 in the figure 1 and 2. Similarly the g4QNNx0 is changed into PA1-V.bas in the figure 1 and 2. In addition, we also included the Uniprot KB accession number of Ves a 1 and PA1-V.bas in the written text and figure caption.
-Figure 2: Resolution needs to be improved.
Response: We have newly constructed the Figure 2 with the 300dpi standard.
-Figure 3: Resolution needs to be improved.
Response: We have newly constructed the Figure 3 with the 300dpi standard and merged with the Ramachandran plot.
-Figure 6: Resolution needs to be improved. And again, 4QNN and Ves a 1 require to be used consistently in the legend and in the figure.
Response: We have newly created the Figure6 with 300dpi resolution. We also used the name Ves a 1 and PA1-V.bas similar to Figure 1 and Figure 2.
-Is it Figure 8 before Figure 9? Should it have Figure number?
Response: Sorry for our mistake. We have added the Figure 8 in the revised manuscript.
- Is the catalytic site of Ves a 1 (Vespa affinis ) conserved with other PLA1?
Response: The answer of your question is “Yes”. Thanks for pointing out this question. We have done sequence alignment of vespid phospholipase A1 among other species and also rat and human lipases. We found that the catalytic site of Ves a 1 is truly conserved with other PLA1 and some lipases. The result could consolidate our finding and discussion.
-Why do author choose DMPC as a possible binding candidate?
Response: Thanks for letting us clarify this issue.We have choose the DMPC as a possible binding candidate because in many previous membrane studies, DMPC was used as to represent the membrane environment. In addition, DMPC is acceptable and common model for the modeling of mammalian phospholipid bilayer. We have pointed out this DMPC as a chosen candidate with the references in the revised manuscript.
-Can Ves a 1 of Vespa affinis also hydrolyze other lipids? Please discuss about this issue.
Response: if we included only phospholipid, Ves a 1 can hydrolyze it because Ves a 1 is specific for phospholipid molecule. For just lipid molecule, the answer is not yet determined according to the review of phospholipase a1. We have added this point into the discussion part too.
As aforementioned, we have tried to fulfill as much as possible your valuable comments. Thanks very much for your guidance.
Kind regards,
Reviewer 2 Report
The manuscript has been improved according to the reviwers’ suggestions. But still the manuscript needs corrections before publication.
Overall
The resolution of the figures is low. Please replace the figures which have high resolution.
Figure 1
The numbers in the figure legend should be used the generalized numbers, such as PDB number and GenBank number. Why did authors use unique numbers (576011171, g4QNNx0) which are specific in this manuscript? And the number in the figure (576011171) is different from the number in the legend (57601171).
Figures 2 and 3
I could not find the difference between figure 2 and figure 3. Please delete one of the figures.
Numbering of the amino acid sequence in figures 2 and 3 is not in accordance with that of description in the manuscript. Please unify the numbering.
Discussion
As you know, vespid phospholipase A1 has some similarities with some lipases. Is the novel lipid binding region of vespid phospholipase A1 conserved in the several lipases? Or the conservation is limited to the Hymenoptera venom phospholipase A1? Please describe about this matter in discussion section.
Author Response
Dear Reviewer,
Thanks for your comments for our manuscript. We would like to address the response for your comment as followed.
-Overall
The resolution of the figures is low. Please replace the figures which have high resolution.
Response: Thanks very much form this point. We have newly created all figures with the 300dpi resolution.
-Figure 1
The numbers in the figure legend should be used the generalized numbers, such as PDB number and GenBank number. Why did authors use unique numbers (576011171, g4QNNx0) which are specific in this manuscript? And the number in the figure (576011171) is different from the number in the legend (57601171).
Response: We do agree with your point and it needs to be improved with the standard code as you said. Thanks for your comment. We do agree that it could make readers confuse. Therefore, in figure 1, we have changed the name of g4QNNx0 and 567011171 into more general name. To eliminate reader’s confusion, the 567011171 is changed into Ves a 1 in the figure 1 and 2. Similarly the g4QNNx0 is changed into PA1-V.bas in the figure 1 and 2. In addition, we also included the Uniprot KB accession number of Ves a 1 and PA1-V.bas in the written text and figure caption so that the reader can track the sequence information source.
-Figures 2 and 3
I could not find the difference between figure 2 and figure 3. Please delete one of the figures.
Numbering of the amino acid sequence in figures 2 and 3 is not in accordance with that of description in the manuscript. Please unify the numbering.
Response: We have merged the figure 3 with Ramachandran plot and we also newly created Figure 2 separately and more friendly to read for the audience. One more thing, we found that we mistakenly put the wrong sequence number in the manuscript. So we checked them again with Uniprot Kb database and in this revised manuscript, we all corrected the numbering of amino acid corresponding the protein sequence of Ves a 1 in both species. The numbering system of amino acid in all figures is consistent throughout the revised manuscript.
-Discussion
As you know, vespid phospholipase A1 has some similarities with some lipases. Is the novel lipid binding region of vespid phospholipase A1 conserved in the several lipases? Or the conservation is limited to the Hymenoptera venom phospholipase A1? Please describe about this matter in discussion section.
Response: Thanks for your suggestion. This can consolidate our findings. We have done sequence alignment of catalytic site and novel lipid binding site region of vespid phospholipase A1. We found that this conversation is not only limited to the Hymenoptera venom PLA1. The novel binding region showed the similarity to human lipases and rat lipases clearly. It informed the novel binding region conserved in some mammalians. Therefore we have added the figure 10 regarding the similarity and wrote this discussion part in the revised manuscript.
As above-mentioned statement, the manuscript was improved from the previous version and we have tried to fill the gap of our research as much as we can.
Thanks very much for reading and giving the constructive comments for our manuscript.
Kind regards,